# Innovative Models Built Based on Image Textures Using Traditional Machine Learning Algorithms for Distinguishing Different Varieties of Moroccan Date Palm Fruit (*Phoenix dactylifera* L.)

Younés Noutfia [1],* and Ewa Ropelewska [2],*

1. Agri Food and Quality Department, National Institute of Agricultural Research, Avenue Ennasr, BP 415 Rabat Principale, Rabat 10090, Morocco
2. Fruit and Vegetable Storage and Processing Department, The National Institute of Horticultural Research, Konstytucji 3 Maja 1/3, 96-100 Skierniewice, Poland
* Correspondence: younes.noutfia@inra.ma (Y.N.); ewa.ropelewska@inhort.pl (E.R.)

**Abstract:** The aim of this study was to develop the procedure for the varietal discrimination of date palm fruit using image analysis and traditional machine learning techniques. The fruit images of 'Mejhoul', 'Boufeggous', 'Aziza', 'Assiane', and 'Bousthammi' date varieties, converted to individual color channels, were processed to extract the texture parameters. After performing the attribute selection, the textures were used to build models intended for the discrimination of different varieties of date palm fruit using machine learning algorithms from Functions, Bayes, Lazy, Meta, and Trees groups. Models were developed for combining image textures selected from a set of all color channels and for sets of textures selected for individual color spaces and color channels. The models, including combined textures selected from all color channels, distinguished all five varieties with an average accuracy reaching 98%, and 'Bousthammi' and 'Mejhoul' were completely correctly discriminated for the SMO (Functions) and IBk (Lazy) machine learning algorithms. By reducing the number of varieties, the correctness of the date palm fruit classification increased. The models developed for the three most different date palm fruit varieties 'Boufeggous', 'Bousthammi', and 'Mejhoul' revealed an average discrimination accuracy of 100% for each algorithm used (SMO, Naive Bayes (Bayes), IBk, LogitBoost (Meta), and LMT (Trees)). In the case of individual color spaces and channels, the accuracies were lower, reaching 97.3% for color space RGB and SMO and LMT algorithms for all five varieties and 99.63% for Naive Bayes and IBk for the 'Boufeggous', 'Bousthammi', and 'Mejhoul' date palm fruits. The results can be used in practice to develop vision systems for sorting and distinguishing the varieties of date palm fruit to authenticate the variety of the fruit intended for further processing.

**Keywords:** date palm; fruit image processing; varietal discrimination; performance metrics

## 1. Introduction

Date palm (*Phoenix dactylifera* L.) belonging to the Arecaceae family is one of the oldest cultivated plants and was domesticated by 3000–4500 BCE in Mesopotamia [1]. Date palm is a socioeconomically important crop, mainly in the Middle East and North Africa. Date palm fruit is rich in carbohydrates, fats, proteins, minerals, fats, and vitamins. Dates can be consumed fresh or used in the food processing industry to produce jam, jelly, juice, syrup, and dried forms. There are more than 2000 date varieties worldwide that differ in their characteristics [2], and in Morocco, we count more than 400 cultivars and autochthonous varieties such as Mejhoul, Boufeggous, Aziza, Najda, Bousekri, and Jihel [3]. Date palm fruit is also a good source of polyphenols, carotenoids, and flavonoids. The antioxidant phytochemicals can be partly responsible for health benefits [4]. Therefore, besides its medical value, the date palm fruit is characterized by functional and nutritional values [5]. There are many varieties of dates marketed at various price ranges. Date varieties may

be of different quality. For example, the flesh and seed weight and dimensions of dates may depend on the variety. Additionally, the color difference of the fruit may be related to genetic variations [6]. Date palm fruit varieties can also differ in the antinutrients and chemical composition, including the content of sugars, micronutrients, amino acids, and protein. These differences can depend on the geographical region, cultivation conditions, chemical fertilization, seed nature, or pre- and post-harvesting treatment techniques [7,8]. Different varieties of date palm fruit can be characterized by different appearance, taste, and nutritional value [9], as well as growth and production including the vulnerability to crop losses caused, among others, by various abiotic and biotic stresses resulting in fruit drop [10].

Due to the fact that some date palm fruit varieties may be characterized by similar morphological features, their taxonomical classification can be complicated. Therefore, biochemical and molecular analyses may be desired to identify date palm fruit varieties [11]. These methods may require complex sample preparation, expensive equipment and chemical reagents, and can be time consuming and labor intensive. Thus, an approach involving a non-destructive, rapid, objective, and inexpensive procedure is needed. Machine vision can meet these demands. It can enable the quantitative analysis of qualitative criteria due to image processing and pattern recognition [12]. For example, combining the color properties extracted from date palm fruit digital images and their processing and statistical analysis can provide a recognition system for date palm fruit variety [12]. In addition to color, the size, shape, and texture of date palm fruit can be computed based on digital images and used to classify fruit into various groups, e.g., considering ripeness [13]. Koklu et al. [14] used morphological, shape, and color parameters extracted from images combined with machine learning models using an artificial neural network (ANN) and logistic regression (LR) to successfully classify date palm fruit varieties. Generally, machine learning has been applied to many fruit studies, including the identification, varietal classification, grading, and sorting of fruits [15]. The efficiency of fruit classification depends, among others, on the usefulness of extracted attributes and the selection of algorithms [16]. In our previous studies, traditional machine learning algorithms were used, for example, for the cultivar classification of fruits [17,18] and quality monitoring of processed or stored fruit [19,20].

In view of the available literature data, the application of image analysis and artificial intelligence can be promising for distinguishing date palm fruit belonging to different varieties and characterized by different developmental stages. The aim of this study was to develop innovative models for the varietal discrimination of Moroccan date palm fruit using image analysis and traditional machine learning techniques. The fruit images of the 'Mejhoul', 'Boufeggous', 'Assiane', 'Aziza', and 'Bousthammi' date varieties converted to individual color channels were processed to extract the texture parameters. The great novelty of this study compared with the literature data involved the development of classification models based on attributes selected from a set of 2160 texture parameters extracted from images in 12 color channels using various machine learning algorithms from the Functions, Bayes, Lazy, Meta, and Trees groups. The use of such a big set of image texture parameters and algorithms from different groups was an innovative approach to distinguishing between the 'Mejhoul', 'Boufeggous', 'Assiane', 'Aziza', and 'Bousthammi' date palm fruit varieties.

## 2. Materials and Methods

### 2.1. Materials

The research material consisted of date palm fruit belonging to five varieties, namely, 'Mejhoul', 'Boufeggous', 'Aziza', 'Assiane', and 'Bousthammi', sampled in 2021 from a cold unit in the southeast of Morocco at the locality of Erfoud (31° 26′ 20″ N, 4° 14′ 37″ W).

Ninety whole undamaged fruits of each variety at the Tamar stage of maturity were used in this study. Fresh batches of date palm fruit of each variety were distributed into cardboard boxes and stored for 24 h at +4 °C at a laboratory before image acquisition. In total, the experimental set included 450 fruits intended for imaging.

### 2.2. Image Acquisition and Processing

The date palm fruits were imaged using a cell phone camera (Samsung Galaxy S10+, Samsung Group, Suwon, Republic of Korea). The cell phone camera was characterized by optical image stabilization, f/1.9 aperture, and optical zoom. The distance between the camera and the sample was 300 mm. The images were acquired in a box on a white background, which was then changed to black to facilitate image processing. In the case of each date palm fruit variety, five images with 18 fruits in each image were obtained. The acquired images were converted to BMP format. Then, the images were processed using MaZda software (Łódź University of Technology, Institute of Electronics, Lodz, Poland). MaZda is a software package for image texture analysis that allows image segmentation, a region of interest (ROI) determination, and the computation of texture parameters from different color channels [21–23].

In the first step, the images were converted to color channels *L* (lightness component from black to white), *a* (color component—green for negative and red for positive values), *b* (color component—blue for negative and yellow for positive values), *R* (red), *G* (green), *B* (blue), *U* (component with color information), *V* (value), *S* (saturation), *X* (component with color information), *Y* (lightness), and *Z* (component with color information). The sample images of date palm fruit varieties are presented in Figure 1. Then, the image segmentation into fruits and the black background was carried out using the brightness threshold. The black background had a pixel brightness intensity equal to 0 and the fruit samples were lighter with a pixel brightness intensity greater than 0. Thus, the lighter date palm fruits were separated from the background and each whole fruit was considered as one ROI. In the case of each color channel, 180 textures based on the histogram, run-length matrix, co-occurrence matrix, autoregressive model, and gradient map were computed. In total, 2160 texture parameters including 180 for each of the 12 color channels (*L*, *a*, *b*, *R*, *G*, *B*, *U*, *V*, *S*, *X*, *Y*, and *Z*) were determined for each fruit.

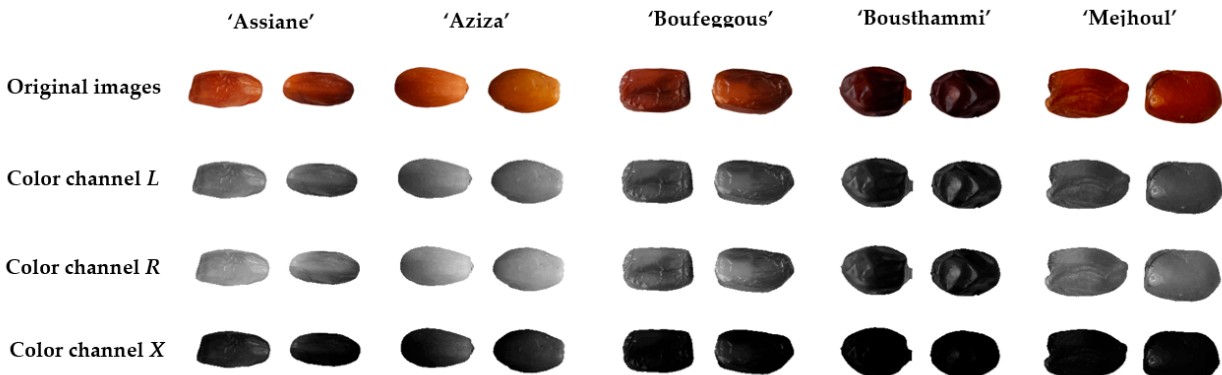

**Figure 1.** Exemplary original images of date palm fruits belonging to different varieties and images in selected color channels.

### 2.3. Varietal Discrimination of Date Palm Fruits

The texture attributes were used to develop the models for the discrimination of different varieties of date palm fruits. A discriminant analysis was carried out using WEKA 3.9 machine learning software (Machine Learning Group, University of Waikato, Hamilton, New Zealand) [24–26]. The texture selection was carried out using the Ranker method with the OneR attribute evaluator. The date palm fruits belonging to different varieties were discriminated using a 10-fold cross-validation mode. This mode randomly divided the dataset into 10 parts. Nine parts were considered as the training sets and the tenth part was the test set. The learning process was performed a total of 10 times on different training sets. Each part was treated as the test set and the remaining as the training sets in turn. The overall error was estimated as the average of 10 error estimates [24]. The algorithms from the Functions, Bayes, Lazy, Meta, and Trees groups were applied. The algorithms providing the most correct results were selected to be presented in the Results

and Discussion section. The first models were built for combining image textures selected from a set of all color channels *L*, *a*, *b*, *R*, *G*, *B*, *U*, *V*, *S*, *X*, *Y*, and *Z*. The discrimination for all five date palm fruit varieties 'Assiane', 'Aziza', 'Boufeggous', 'Bousthammi', and 'Mejhoul' was performed. Additionally, the analysis was carried out for three selected varieties to check whether reducing the number of varieties affects the correctness of the classification. The next models were developed for sets of textures selected for each color space and channel separately. The results for the selected channels providing the most successful discriminations are presented in this paper. As the results of discriminant analysis, the confusion matrices, average accuracies, and the values of TP Rate (true positive rate), FP Rate (false positive rate), Precision, Recall, F-Measure, MCC (Matthews correlation coefficient), ROC Area (receiver operating characteristic area), and PRC Area (precision–recall area) were determined [27–30].

## 3. Results and Discussion

In the case of discrimination performed for all five date palm fruit varieties using image textures selected from all color channels, very satisfactory results were obtained. The confusion matrices and average accuracies are presented in Table 1.

**Table 1.** Confusion matrices and average accuracies of varietal discrimination of date palm fruits for the models built based on a set of image textures selected from color channels *L*, *a*, *b*, *R*, *G*, *B*, *U*, *V*, *S*, *X*, *Y*, and *Z*.

| Algorithm | Predicted Class (%) | | | | | Actual Class | Average Accuracy (%) |
|---|---|---|---|---|---|---|---|
| | 'Assiane' | 'Aziza' | 'Boufeggous' | 'Bousthammi' | 'Mejhoul' | | |
| SMO (Functions) | 97 | 3 | 0 | 0 | 0 | 'Assiane' | 98 |
| | 7 | 93 | 0 | 0 | 0 | 'Aziza' | |
| | 1 | 0 | 99 | 0 | 0 | 'Boufeggous' | |
| | 0 | 0 | 0 | 100 | 0 | 'Bousthammi' | |
| | 0 | 0 | 0 | 0 | 100 | 'Mejhoul' | |
| Naive Bayes (Bayes) | 88 | 8 | 4 | 0 | 0 | 'Assiane' | 95 |
| | 9 | 91 | 0 | 0 | 0 | 'Aziza' | |
| | 3 | 0 | 97 | 0 | 0 | 'Boufeggous' | |
| | 0 | 0 | 0 | 100 | 0 | 'Bousthammi' | |
| | 0 | 0 | 0 | 0 | 100 | 'Mejhoul' | |
| IBk (Lazy) | 91 | 3 | 6 | 0 | 0 | 'Assiane' | 98 |
| | 2 | 98 | 0 | 0 | 0 | 'Aziza' | |
| | 1 | 0 | 99 | 0 | 0 | 'Boufeggous' | |
| | 0 | 0 | 0 | 100 | 0 | 'Bousthammi' | |
| | 0 | 0 | 0 | 0 | 100 | 'Mejhoul' | |
| LogitBoost (Meta) | 88 | 7 | 5 | 0 | 0 | 'Assiane' | 95 |
| | 6 | 94 | 0 | 0 | 0 | 'Aziza' | |
| | 6 | 0 | 94 | 0 | 0 | 'Boufeggous' | |
| | 0 | 0 | 0 | 100 | 0 | 'Bousthammi' | |
| | 0 | 0 | 0 | 0 | 100 | 'Mejhoul' | |
| LMT (Trees) | 93 | 4 | 3 | 0 | 0 | 'Assiane' | 97 |
| | 8 | 92 | 0 | 0 | 0 | 'Aziza' | |
| | 2 | 0 | 98 | 0 | 0 | 'Boufeggous' | |
| | 0 | 0 | 0 | 100 | 0 | 'Bousthammi' | |
| | 0 | 0 | 0 | 0 | 100 | 'Mejhoul' | |

It was found that the discrimination metrics depended on the machine learning algorithm. A high average accuracy equal to 98% was obtained for the SMO (group of Functions) and IBk (group of Lazy) algorithms. However, the confusion matrices were slightly different. In the case of both algorithms (SMO and IBk), the 'Bousthammi' and 'Mejhoul' date palm fruit varieties were correctly discriminated 100% of the time. It meant that there was a set of fruit image textures to completely correctly distinguish these varieties from each other and from other varieties. Moreover, no fruits belonging to other

varieties were incorrectly classified as 'Bousthammi' or 'Mejhoul'. These results are very promising. For the SMO and IBk algorithms, the 'Boufeggous' date palm fruits were correctly discriminated in 99% of cases, and the remaining 1% of the fruit was incorrectly included in the predicted class 'Assiane'. The lowest accuracies and a high mixing of cases were observed for the 'Assiane' and 'Aziza' varieties. In the case of the SMO, 'Assiane' fruits were correctly distinguished 97% of the time and the remaining 3% were classified as 'Aziza'. The 'Aziza' variety was characterized by a discrimination accuracy of 93%, and 7% of cases belonging to the actual class 'Aziza' were incorrectly included in the predicted class 'Assiane'. In the case of the IBk algorithm, the accuracy of 91% was obtained for 'Assiane' (the remaining 3% were classified as 'Aziza' and 6% as 'Boufeggous'), and 98% was determined for 'Aziza' (the remaining 2% were classified as 'Assiane'). The models built using other algorithms also provided high average accuracies equal to 97% for LMT (group of Trees), 95% for Naive Bayes (group of Bayes) and LogitBoost (group of Meta). In the case of each algorithm, the 'Bousthammi' and 'Mejhoul' fruits were discriminated with a correctness of 100%. Moreover, 'Boufeggous' fruits were distinguished from other varieties with very high accuracies equal to 98% for LMT, 97% for Naive Bayes, and 94% for LogitBoost. The 'Assiane', and 'Aziza' date palm fruits were discriminated with the lowest accuracies, and high mixing of cases between these varieties was revealed. The lowest accuracies equal to 88% and 91% for 'Assiane', and 'Aziza', respectively, were found for the model built using the Naive Bayes algorithm. As much as 8% of cases belonging to 'Assiane' were classified as 'Aziza', and the remaining 4% as 'Boufeggous', whereas 9% of cases from the actual class 'Aziza' were classified as 'Assiane'.

The other discrimination performance metrics, such as TP Rate, FP Rate, Precision, Recall, F-Measure, MCC, ROC Area, and PRC Area, for the models built based on a set of textures selected from all color channels are shown in Table 2. The most satisfactory results were obtained for the 'Bousthammi' and 'Mejhoul' date palm fruits. In the case of models developed using each algorithm (SMO, Naive Bayes, IBk, LogitBoost, and LMT), the values of TP Rate, Precision, Recall, F-Measure, MCC, ROC Area, and PRC Area were equal to 1.000 and the FP Rate was 0.000. This meant that the 'Bousthammi' and 'Mejhoul' varieties were completely correctly distinguished from each other and from other varieties.

By reducing the number of varieties, the correctness of the date palm fruit classification increased. The three most different 'Boufeggous', 'Bousthammi', and 'Mejhoul' varieties were discriminated with an average accuracy of 100% in the case of each of the applied algorithms (SMO, Naive Bayes, IBk, LogitBoost, LMT) (Table 3). This meant that the fruit images of these varieties were completely different in terms of the selected textures.

The completely correct discrimination of date palm fruits was confirmed by the values of TP Rate, Precision, Recall, F-Measure, MCC, ROC Area, and PRC Area (1.000) and FP Rate (0.000) for each variety and algorithm (Table 4).

The obtained results were very successful and proved that it is possible to completely distinguish date palm fruit varieties based on image textures selected from all color channels. The application of image analysis and machine learning allowed for distinguishing date palm fruit varieties with an accuracy reaching 100%. Date palm fruit varieties can also be successfully discriminated using, among others, a metabolomics approach [5]. Khalil et al. [31] reported that varietal differences were more noticeable in terms of the number of volatiles rather than their composition and 2,3-butanediol, cinnamaldehyde, hexanol, and hexanal had the greatest impact on the classification of different date palm varieties determined using multivariate data analyses. Farag et al. [32] revealed that sugars and flavonols also contributed to date palm fruit varietal classification performed using principal component and clustering analyses. The varietal discrimination of date palm based on criteria related to the vegetative system before fruiting (leaf, pinnae, and spine characteristics) using PCA (principal component analysis) and hierarchical ascendant, UPGMA (unweighted pair-group method arithmetic average) was performed by Elhoumaizi et al. [33]. Extensive chemometric methods involving PCA and PLS-DA (leave-one-out cross-validation model) coupled with NIRS (near-infrared reflectance spectroscopy)

and FTIR/ATR (Fourier-transform infrared/attenuated total reflectance) and NMR (nuclear magnetic resonance) spectroscopy were used for the identification of the sex differentiation in immature date palm leaves [34]. The morphological characteristics of date palm fruit can be considered for varietal identification. However, narrow distinguishing morphological parameters cause difficulties and may result in the need for confirmation of the variety using genetic evidence [35,36]. Additionally, morphological markers can depend on the environmental conditions and developmental stage. Therefore, they can be unreliable for varietal identification and many molecular markers were developed for the analysis of the genetic diversity of date palm fruit [37]. Among others, the genetic diversity of date palm was identified by Mathew et al. [38] using molecular genome-wide SNP (single nucleotide polymorphism) analysis based on DNA sequences and by Srivashtav et al. [39] using RAPD (random amplified polymorphic DNA) and ISSR (inter simple sequence repeats) markers. As reported in the available literature, there are many effective methods and techniques for identifying a variety of date palm fruit. However, our own results are a great novelty in the non-destructive differentiation of date varieties. The classification models including image textures selected from color channels $L$, $a$, $b$, $R$, $G$, $B$, $U$, $V$, $S$, $X$, $Y$, and $Z$ were innovative. Traditional machine learning algorithms turned out to be very successful in developing models to distinguish date palm varieties. This prompts further research to discriminate different varieties of date palm fruit, for example using deep learning.

**Table 2.** Performance metrics of varietal discrimination of date palm fruits for the models built based on a set of image textures selected from color channels $L$, $a$, $b$, $R$, $G$, $B$, $U$, $V$, $S$, $X$, $Y$, and $Z$.

| Algorithm | Class | TP Rate | FP Rate | Precision | Recall | F-Measure | MCC | ROC Area | PRC Area |
|---|---|---|---|---|---|---|---|---|---|
| SMO (Functions) | 'Assiane' | 0.967 | 0.019 | 0.926 | 0.967 | 0.946 | 0.932 | 0.984 | 0.908 |
| | 'Aziza' | 0.933 | 0.008 | 0.966 | 0.933 | 0.949 | 0.937 | 0.989 | 0.937 |
| | 'Boufeggous' | 0.989 | 0.000 | 1.000 | 0.989 | 0.994 | 0.993 | 0.997 | 0.992 |
| | 'Bousthammi' | 1.000 | 0.000 | 1.000 | 1.000 | 1.000 | 1.000 | 1.000 | 1.000 |
| | 'Mejhoul' | 1.000 | 0.000 | 1.000 | 1.000 | 1.000 | 1.000 | 1.000 | 1.000 |
| Naive Bayes (Bayes) | 'Assiane' | 0.878 | 0.031 | 0.878 | 0.878 | 0.878 | 0.847 | 0.988 | 0.942 |
| | 'Aziza' | 0.911 | 0.019 | 0.921 | 0.911 | 0.916 | 0.895 | 0.994 | 0.981 |
| | 'Boufeggous' | 0.967 | 0.011 | 0.956 | 0.967 | 0.961 | 0.952 | 0.999 | 0.994 |
| | 'Bousthammi' | 1.000 | 0.000 | 1.000 | 1.000 | 1.000 | 1.000 | 1.000 | 1.000 |
| | 'Mejhoul' | 1.000 | 0.000 | 1.000 | 1.000 | 1.000 | 1.000 | 1.000 | 1.000 |
| IBk (Lazy) | 'Assiane' | 0.911 | 0.008 | 0.965 | 0.911 | 0.937 | 0.923 | 0.951 | 0.897 |
| | 'Aziza' | 0.978 | 0.008 | 0.967 | 0.978 | 0.972 | 0.965 | 0.985 | 0.950 |
| | 'Boufeggous' | 0.989 | 0.014 | 0.947 | 0.989 | 0.967 | 0.959 | 0.988 | 0.939 |
| | 'Bousthammi' | 1.000 | 0.000 | 1.000 | 1.000 | 1.000 | 1.000 | 1.000 | 1.000 |
| | 'Mejhoul' | 1.000 | 0.000 | 1.000 | 1.000 | 1.000 | 1.000 | 1.000 | 1.000 |
| LogitBoost (Meta) | 'Assiane' | 0.878 | 0.028 | 0.888 | 0.878 | 0.883 | 0.854 | 0.989 | 0.954 |
| | 'Aziza' | 0.944 | 0.017 | 0.934 | 0.944 | 0.939 | 0.924 | 0.996 | 0.987 |
| | 'Boufeggous' | 0.944 | 0.017 | 0.934 | 0.944 | 0.939 | 0.924 | 0.997 | 0.988 |
| | 'Bousthammi' | 1.000 | 0.000 | 1.000 | 1.000 | 1.000 | 1.000 | 1.000 | 1.000 |
| | 'Mejhoul' | 1.000 | 0.000 | 1.000 | 1.000 | 1.000 | 1.000 | 1.000 | 1.000 |
| LMT (Trees) | 'Assiane' | 0.933 | 0.025 | 0.903 | 0.933 | 0.918 | 0.897 | 0.992 | 0.970 |
| | 'Aziza' | 0.922 | 0.008 | 0.965 | 0.922 | 0.943 | 0.930 | 0.996 | 0.984 |
| | 'Boufeggous' | 0.978 | 0.008 | 0.967 | 0.978 | 0.972 | 0.965 | 1.000 | 0.998 |
| | 'Bousthammi' | 1.000 | 0.000 | 1.000 | 1.000 | 1.000 | 1.000 | 1.000 | 1.000 |
| | 'Mejhoul' | 1.000 | 0.000 | 1.000 | 1.000 | 1.000 | 1.000 | 1.000 | 1.000 |

TP Rate—true positive rate; FP Rate—false positive rate; MCC—Matthews correlation coefficient; ROC Area—receiver operating characteristic area; PRC Area—precision–recall area.

**Table 3.** Confusion matrices and average accuracies of discrimination of 'Boufeggous', 'Bousthammi', and 'Mejhoul' date palm fruits for the models built based on a set of image textures selected from color channels $L$, $a$, $b$, $R$, $G$, $B$, $U$, $V$, $S$, $X$, $Y$, and $Z$.

| Algorithm | Predicted Class (%) | | | Actual Class | Average Accuracy (%) |
|---|---|---|---|---|---|
| | 'Boufeggous' | 'Bousthammi' | 'Mejhoul' | | |
| SMO (Functions) | 100 | 0 | 0 | 'Boufeggous' | |
| | 0 | 100 | 0 | 'Bousthammi' | 100 |
| | 0 | 0 | 100 | 'Mejhoul' | |

**Table 3.** *Cont.*

| Algorithm | Predicted Class (%) | | | Actual Class | Average Accuracy (%) |
|---|---|---|---|---|---|
| | 'Boufeggous' | 'Bousthammi' | 'Mejhoul' | | |
| Naive Bayes (Bayes) | 100 | 0 | 0 | 'Boufeggous' | |
| | 0 | 100 | 0 | 'Bousthammi' | 100 |
| | 0 | 0 | 100 | 'Mejhoul' | |
| IBk (Lazy) | 100 | 0 | 0 | 'Boufeggous' | |
| | 0 | 100 | 0 | 'Bousthammi' | 100 |
| | 0 | 0 | 100 | 'Mejhoul' | |
| LogitBoost (Meta) | 100 | 0 | 0 | 'Boufeggous' | |
| | 0 | 100 | 0 | 'Bousthammi' | 100 |
| | 0 | 0 | 100 | 'Mejhoul' | |
| LMT (Trees) | 100 | 0 | 0 | 'Boufeggous' | |
| | 0 | 100 | 0 | 'Bousthammi' | 100 |
| | 0 | 0 | 100 | 'Mejhoul' | |

**Table 4.** Performance metrics of discrimination of 'Boufeggous', 'Bousthammi', and 'Mejhoul' date palm fruits for the models built based on a set of image textures selected from color channels *L*, *a*, *b*, *R*, *G*, *B*, *U*, *V*, *S*, *X*, *Y*, and *Z*.

| Algorithm | Class | TP Rate | FP Rate | Precision | Recall | F-Measure | MCC | ROC Area | PRC Area |
|---|---|---|---|---|---|---|---|---|---|
| SMO (Functions) | 'Boufeggous' | 1.000 | 0.000 | 1.000 | 1.000 | 1.000 | 1.000 | 1.000 | 1.000 |
| | 'Bousthammi' | 1.000 | 0.000 | 1.000 | 1.000 | 1.000 | 1.000 | 1.000 | 1.000 |
| | 'Mejhoul' | 1.000 | 0.000 | 1.000 | 1.000 | 1.000 | 1.000 | 1.000 | 1.000 |
| Naive Bayes (Bayes) | 'Boufeggous' | 1.000 | 0.000 | 1.000 | 1.000 | 1.000 | 1.000 | 1.000 | 1.000 |
| | 'Bousthammi' | 1.000 | 0.000 | 1.000 | 1.000 | 1.000 | 1.000 | 1.000 | 1.000 |
| | 'Mejhoul' | 1.000 | 0.000 | 1.000 | 1.000 | 1.000 | 1.000 | 1.000 | 1.000 |
| IBk (Lazy) | 'Boufeggous' | 1.000 | 0.000 | 1.000 | 1.000 | 1.000 | 1.000 | 1.000 | 1.000 |
| | 'Bousthammi' | 1.000 | 0.000 | 1.000 | 1.000 | 1.000 | 1.000 | 1.000 | 1.000 |
| | 'Mejhoul' | 1.000 | 0.000 | 1.000 | 1.000 | 1.000 | 1.000 | 1.000 | 1.000 |
| LogitBoost (Meta) | 'Boufeggous' | 1.000 | 0.000 | 1.000 | 1.000 | 1.000 | 1.000 | 1.000 | 1.000 |
| | 'Bousthammi' | 1.000 | 0.000 | 1.000 | 1.000 | 1.000 | 1.000 | 1.000 | 1.000 |
| | 'Mejhoul' | 1.000 | 0.000 | 1.000 | 1.000 | 1.000 | 1.000 | 1.000 | 1.000 |
| LMT (Trees) | 'Boufeggous' | 1.000 | 0.000 | 1.000 | 1.000 | 1.000 | 1.000 | 1.000 | 1.000 |
| | 'Bousthammi' | 1.000 | 0.000 | 1.000 | 1.000 | 1.000 | 1.000 | 1.000 | 1.000 |
| | 'Mejhoul' | 1.000 | 0.000 | 1.000 | 1.000 | 1.000 | 1.000 | 1.000 | 1.000 |

TP Rate—true positive rate; FP Rate—false positive rate; MCC—Matthews correlation coefficient; ROC Area—receiver operating characteristic area; PRC Area—precision–recall area.

## 4. Conclusions

The obtained results revealed the usefulness of the innovative models developed based on selected image features using traditional machine learning algorithms to classify date palm varieties that can be used in practice to distinguish and authenticate varieties before consumption and processing. Five date varieties, namely, 'Assiane', 'Aziza', 'Boufeggous', 'Bousthammi', and 'Mejhoul', were distinguished with an average accuracy reaching 98% using the SMO (group of Functions) and IBk (group of Lazy) algorithms. The models built to classify three selected varieties ('Boufeggous', 'Bousthammi', and 'Mejhoul') produced a correctness level of 100% using the SMO (Functions), Naive Bayes (Bayes), IBk (Lazy), LogitBoost (Meta), and LMT (Trees). The most successful models included a set of combined image textures selected from color channels *L*, *a*, *b*, *R*, *G*, *B*, *U*, *V*, *S*, *X*, *Y*, and *Z*. In the case of models built separately based on image textures from selected color spaces and color channels, the accuracies of classification were slightly lower. Further research may be expanded to use deep learning to distinguish date palm fruit varieties with high correctness.

**Author Contributions:** Conceptualization, E.R. and Y.N.; methodology, E.R. and Y.N.; software, E.R.; validation, E.R. and Y.N.; formal analysis, E.R. and Y.N.; writing—original draft preparation, E.R. and Y.N.; writing—review and editing, E.R. and Y.N.; supervision, E.R. All authors have read and agreed to the published version of the manuscript.

**Funding:** This work was performed in the frame of the POLONEZ BIS 2 financed by the National Science Centre with the registration number 2022/45/P/NZ9/03904. Project title: "A novel approach to the assessment of date fruit quality (*Phoenix dactylifera* L.) under different storage conditions, using innovative models based on image analyses and machine learning" (M-LEARN4DATE). POLONEZ BIS 2 has received funding from the European Union's Horizon 2020 research and innovation program under the Marie Skłodowska-Curie Grant Agreement concluded between the National Science Centre and the European Commission no. 945339.

**Institutional Review Board Statement:** Not applicable.

**Data Availability Statement:** The data presented in this study are available upon request from the corresponding author.

**Conflicts of Interest:** The authors declare no conflict of interest.

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
