# Peer review of "Innovative Models Built Based on Image Textures Using Traditional Machine Learning Algorithms for Distinguishing Different Varieties of Moroccan Date Palm Fruit (Phoenix dactylifera L.)"

_agriculture, doi:10.3390/agriculture13010026_

Round 1

Reviewer 1 Report

Overall, this was an interesting submission to review. The writing (English) is understandable but needs some minor copy editing to clean up a few things. The paper should be revised with some more detail in the materials and methods section, and the introduction can be shortened, particularly 33-57. Most of this information is not relevant to the research. More details are needed in the materials and methods section: growing location, date of harvest, length of storage, storage conditions, type of camera, brief description of what Mazda software is (I found "image texture analysis" on line but it was not apparent from the description in the submission). The main scientific question I have is: were the results validated by blind testing? That is, after the models were derived, could the algorithms distinguish fruits presented without their identity being previously known? If this was the case, it was not apparent to me from the text.

Author Response

Responses to Reviewer 1:

Comment 1. Overall, this was an interesting submission to review. The writing (English) is understandable but needs some minor copy editing to clean up a few things.

Response 1. Thank you very much for your careful reading of the manuscript and this comment. The English writing has been corrected.

Comment 2. The paper should be revised with some more detail in the materials and methods section, and the introduction can be shortened, particularly 33-57. Most of this information is not relevant to the research.

Response 2. We are grateful to the Reviewer for this detailed comment. The introduction has been shortened. The materials and methods section has been supplemented with more detailed information as indicated in the comments below.

Comment 3. More details are needed in the materials and methods section: growing location, date of harvest, length of storage, storage conditions, type of camera, brief description of what Mazda software is (I found "image texture analysis" on line but it was not apparent from the description in the submission).

Response 3. It has been corrected as follows:

“The research material consisted of date palm fruits belonging to five varieties namely ‘Mejhoul’, ‘Boufeggous’, ‘Aziza’, ‘Assiane’, and ‘Bousthammi’, sampled in 2021 from a cold unit in the southeast of Morocco at locality of Erfoud (31° 26′ 20″ north, 4° 14′ 37″ west).

“Ninety whole undamaged fruits of each variety at the Tamar stage of maturity were used in this study. Fresh batches of date palm fruit of each variety were distributed into cardboard boxes and stored for 24 hours at +4°C at a laboratory before image acquisition. In total, the experimental set included 450 fruits intended for imaging.”

“Date palm fruits were imaged using a cell phone camera (Samsung Galaxy S10+, Samsung Group, Suwon, Republic of Korea). The cell phone camera was characterized by optical image stabilization, f/1.9 aperture, and Optical Zoom. The distance of camera from the sample was 300 mm. The images were acquired in a box on a white background, which was then changed to black to facilitate image processing.”

“the images were processed using Mazda software (Łódź University of Technology, Institute of Electronics, Poland). MaZda is a software package for image texture analysis that allows image segmentation, a region of interest (ROI) determination, and the computation of texture parameters from different color channels.”

Comment 4. The main scientific question I have is: were the results validated by blind testing? That is, after the models were derived, could the algorithms distinguish fruits presented without their identity being previously known? If this was the case, it was not apparent to me from the text.

Response 4. It has been explained in more detail in the manuscript as follows:

“The date palm fruits belonging to different varieties were discriminated using a 10-fold cross-validation mode. This mode randomly divided the dataset into 10 parts. Nine parts were considered as the training sets and the tenth part was the test set. The learning process was performed a total of 10 times on different training sets. Each part was treated as the test set and the remaining as the training sets in turn. The overall error was estimated as the average of 10 error estimates. ”

Reviewer 2 Report

The manuscript is written with clear understanding of the project addressed. However, there are major concerns that need to be addressed to enhance the quality of the manuscript. My specific comments are as follows:

Introduction:

Add on literatures on applications of traditional machine learning in fruit classification

Based on your objectives, please compare how your study is different from those that have already been published

Methods:

How about the storage of dates? Any specific room temperatures/packaging materials used? Explain briefly

L88: “Date fruits were images using a digital camera.” Add brand, city, country of the digital camera

What about the camera device settings; for examples: distances of camera from the sample, focal length/zoom range/etc. is it taken in a control environment setting such as in a controlled box

Explain the image segmentation process in details. How the selection of ROI was performed/what method is used; eg thresholding/otsu/etc

What is the split ratio of training and testing dataset. Please justify

Results:

Instead of mentioning the results, the authors should justify/explain the findings

 Conclusion:

Add on main finding/results of the study. What are the main outcome based on the results. The authors should highlighted this matter

Which machine learning algorithm produce the best result

General comments:

Please check the reference styles and grammar of the manuscript.

Author Response

Responses to Reviewer 2:

The manuscript is written with clear understanding of the project addressed. However, there are major concerns that need to be addressed to enhance the quality of the manuscript. My specific comments are as follows:

Introduction:

Comment 1. Add on literatures on applications of traditional machine learning in fruit classification

Response 1.   It has been added as follows:

“Generally, machine learning has been applied to many fruit studies, including identification, varietal classification, grading, and sorting of fruits [15]. The efficiency of fruit classification depends, among others, on the usefulness of extracted attributes and the selection of algorithms [16]. In the previous own studies, traditional machine learning algorithms were used, for example, for cultivar classification of fruits [17,18] and quality monitoring of processed or stored fruit [19,20].” 

Comment 2. Based on your objectives, please compare how your study is different from those that have already been published

Response 2. It has been specified as follows:

“The great novelty of the own study compared with the literature data involved the development of classification models based on attributes selected from a set of 2160 texture parameters extracted from images in 12 color channels using various machine learning algorithms from Functions, Bayes, Lazy, Meta, and Trees groups. The use of such a big set of image texture parameters and algorithms from different groups was an innovative approach to distinguishing ‘Mejhoul’, ‘Boufeggous’, ‘Assiane’, ‘Aziza’, and ‘Bousthammi’ date palm fruit varieties.”

Methods:

Comment 3. How about the storage of dates? Any specific room temperatures/packaging materials used? Explain briefly

Response 3. It has been explained as follows:

“Fresh batches of date palm fruit of each variety were distributed into cardboard boxes and stored for 24 hours at +4°C at a laboratory before image acquisition. In total, the experimental set included 450 fruits intended for imaging.”

Comment 4.      L88: “Date fruits were images using a digital camera.” Add brand, city, country of the digital camera

Response 4. It has been corrected as follows:

 “Date palm fruits were imaged using a cell phone camera (Samsung Galaxy S10+, Samsung Group, Suwon, Republic of Korea).”

Comment 5. What about the camera device settings; for examples: distances of camera from the sample, focal length/zoom range/etc. is it taken in a control environment setting such as in a controlled box

Response 5. It has been corrected as follows:

 “The cell phone camera was characterized by optical image stabilization, f/1.9 aperture, and Optical Zoom. The distance of the camera from the sample was 300 mm. The images were acquired in a box on a white background, which was then changed to black to facilitate image processing.”

Comment 6. Explain the image segmentation process in details. How the selection of ROI was performed/what method is used; eg thresholding/otsu/etc

Response 6. It has been corrected as follows:

“Then, the image segmentation into fruits and the black background was carried out using the brightness threshold. The black background had a pixel brightness intensity equal to 0 and samples were lighter with a pixel brightness intensity greater than 0. Thus, the lighter date palm fruits were separated from the background and each whole fruit was considered as one ROI.”

Comment 7. What is the split ratio of training and testing dataset. Please justify

Response 7.  It has been indicated as follows:

“The date palm fruits belonging to different varieties were discriminated using a 10-fold cross-validation mode. This mode randomly divided the dataset into 10 parts. Nine parts were considered as the training sets and the tenth part was the test set. The learning process was performed a total of 10 times on different training sets. Each part was treated as the test set and the remaining as the training sets in turn. The overall error was estimated as the average of 10 error estimates.”

Results:

Comment 8. Instead of mentioning the results, the authors should justify/explain the findings

Response 8. Some of the data present in Tables have been removed from the text and an explanation has been added instead.

Conclusion:

Comment 9. Add on main finding/results of the study. What are the main outcome based on the results. The authors should highlighted this matter

Response 9. It has been indicated as follows:

“The obtained results revealed the usefulness of the innovative models developed based on selected image features using traditional machine learning algorithms to classify date palm varieties that can be used in practice to distinguish and authenticate varieties before consumption and processing.”

Comment 10. Which machine learning algorithm produce the best result

Response 10. It has been indicated as follows:

“Five date varieties namely ‘Assiane’, ‘Aziza’, ‘Boufeggous’, ‘Bousthammi’, and ‘Mejhoul’ were distinguished with an average accuracy reaching 98% using the SMO (group of Functions) and IBk (group of Lazy) algorithms. Whereas models built to classify three selected varieties (‘Boufeggous’, ‘Bousthammi’, and ‘Mejhoul’) produced correctness of up to 100% using the SMO (Functions), Naive Bayes (Bayes), IBk (Lazy), LogitBoost (Meta), and LMT (Trees).”

General comments:

Comment 11. Please check the reference styles and grammar of the manuscript.

Response 11. The reference styles and grammar have been corrected.

Reviewer 3 Report

The current manuscript entitled “Innovative models built based on image textures using traditional machine learning algorithms for distinguishing different varieties of Moroccan date fruit (Phoenix dactylifera L.)” by Noutfia and Ropelewska deals with the development of vision systems for sorting and distinguishing the varieties of date fruit to authenticate the variety of the fruit intended for further processing. After a careful reading, I found this study well designed, structured, and supported with sufficient data to authenticate the developed date palm fruit sorting and categorization. However, I have pointed out several grey points in the manuscript which need to be justified in the revised version. Thus, I suggest minor revisions. My specific comments are:

1.      Authors are suggested to consistently use “date palm” instead of “date” in the entire manuscript.

2.      The aims and objectives (lines 74-79) should be given in a separate paragraph.

3.      Since the color channels are not defined (abbreviations) in both the abstract and introduction section, I suggest they remove and provide additional information about them under the methods section.

4.      Line 41: it is required to mention some popular varieties of Moroccan date fruit in this sentence.

5.      Line 98: I suspect that 2160 is the sample number (n) and not the total number of texture parameters.

6.      Mention the geocoordinate of the sampling locations (Line 83).

7.      The accuracy parameter should be corrected to three decimal points similar to the performance metrics parameters.

8.      Line 173-185: This text is just a repetition of tabular data; I suggest removing it or writing differently.

Author Response

Responses to Reviewer 3:

The current manuscript entitled “Innovative models built based on image textures using traditional machine learning algorithms for distinguishing different varieties of Moroccan date fruit (Phoenix dactylifera L.)” by Noutfia and Ropelewska deals with the development of vision systems for sorting and distinguishing the varieties of date fruit to authenticate the variety of the fruit intended for further processing. After a careful reading, I found this study well designed, structured, and supported with sufficient data to authenticate the developed date palm fruit sorting and categorization. However, I have pointed out several grey points in the manuscript which need to be justified in the revised version. Thus, I suggest minor revisions. My specific comments are:

Comment 1.      Authors are suggested to consistently use “date palm” instead of “date” in the entire manuscript.

Response 1. Thank you very much for this comment. It has been changed.

Comment 2. The aims and objectives (lines 74-79) should be given in a separate paragraph.

Response 2. Thank you for this comment. It has been corrected.   

Comment 3. Since the color channels are not defined (abbreviations) in both the abstract and introduction section, I suggest they remove and provide additional information about them under the methods section.

Response 3. It has been corrected according to this comment. Abbreviations have been deleted from the abstract and introduction section and explained in the methods section as follows:

“In the first step, the images were converted to color channels L (Lightness component from black to white), a (color component—green for negative and red for positive values), b (color component—blue for negative and yellow for positive values), R (Red), G (Green), B (Blue), U (component with color information), V (Value), S (Saturation), X (component with color information), Y (Lightness), and Z (component with color information).”

Comment 4. Line 41: it is required to mention some popular varieties of Moroccan date fruit in this sentence.

Response 4. It has been corrected as follows:

“in Morocco, we count more than 400 cultivars and autochthonous varieties like Mejhoul, Boufeggous, Aziza, Najda, Bousekri, Jihel…[3].”

  1. Sedra, M.H. Date palm status and perspective in Morocco. In Date palm genetic resources and utilization. Springer, Dordrecht 2015, 257-323.

Comment 5. Line 98: I suspect that 2160 is the sample number (n) and not the total number of texture parameters.

Response 5. It has been explained in more detail as follows:

“In the case of each color channel, 180 textures based on the histogram, run-length matrix, co-occurrence matrix, autoregressive model and gradient map were computed. In total, 2160 texture parameters including 180 for each of the 12 color channels (L, a, b, R, G, B, U, V, S, X, Y, and Z) were determined for each fruit.”

Comment 6. Mention the geocoordinate of the sampling locations (Line 83).

Response 6. It has been corrected as follows:

“The research material consisted of date palm fruits belonging to five varieties namely ‘Mejhoul’, ‘Boufeggous’, ‘Aziza’, ‘Assiane’, and ‘Bousthammi’, sampled in 2021 from a cold unit in the southeast of Morocco at the locality of Erfoud (31° 26′ 20″ north, 4° 14′ 37″ west).

Comment 7. The accuracy parameter should be corrected to three decimal points similar to the performance metrics parameters.

Response 7. Thank you very much for this detailed comment. The accuracy parameter was determined using WEKA 3.9 machine learning software. This software produces the values of the accuracy parameter with two decimal points. Therefore, such values are presented in the manuscript.

Comment 8. Line 173-185: This text is just a repetition of tabular data; I suggest removing it or writing differently.

Response 8.  It has been corrected as follows:

“The other discrimination performance metrics, such as TP Rate, FP Rate, Preci-sion, Recall, F-Measure, MCC, ROC Area, and PRC Area for the models built based on a set of textures selected from all color channels are shown in Table 2. The most satisfactory results were obtained for ‘Bousthammi’ and ‘Mejhoul’ date palm fruits. In the case of models developed using each algorithm (SMO, Naive Bayes, IBk, LogitBoost, and LMT), the values of TP Rate, Precision, Recall, F-Measure, MCC, ROC Area, and PRC Area were equal to 1.000 and FP Rate was 0.000. It meant that ‘Bousthammi’ and ‘Mejhoul’ varieties were completely correctly distinguished from each other and from other varieties.”

Round 2

Reviewer 2 Report

The authors have addressed all the comments. Hence, the paper can be accepted as it is.